# Contributions of Ammonia-Oxidizing Archaea and Bacteria to Nitrous Oxide Production in Intensive Greenhouse Vegetable Fields

Yubing Dong [1,2], Xintong Xu [1], Junqian Zhang [1], Ying Jiao [1], Bingxue Wang [1], Chenyuan Wang [1] and Zhengqin Xiong [1,*]

1   Jiangsu Key Laboratory of Low Carbon Agriculture and GHGs Mitigation, College of Resources and Environmental Sciences, Nanjing Agricultural University, Nanjing 210095, China; dongyubing178@163.com (Y.D.); 2020203058@stu.njau.edu.cn (X.X.); 2021103089@stu.njau.edu.cn (J.Z.); 2022103086@stu.njau.edu.cn (Y.J.); 2022103085@stu.njau.edu.cn (B.W.); 2022803237@stu.njau.edu.cn (C.W.)
2   Huaiyin Institute of Agricultural Sciences of Xuhuai Region in Jiangsu, Jiangsu Academy of Agricultural Sciences, Huaian 223001, China
*   Correspondence: zqxiong@njau.edu.cn; Tel.: +86-25-84395210

**Abstract:** With excessive nitrogen (N) input, high nitrous oxide ($N_2O$) emissions are frequently observed in greenhouse vegetable fields. We hypothesized that the underlying production mechanisms can be derived across a wide selection of vegetable fields in the middle and lower reaches of the Yangtze River. Thus, we investigated the emission characteristics and relative contributions of ammonia-oxidizing archaea (AOA) and bacteria (AOB) and other microbial processes to the $N_2O$ production from five long-term greenhouse vegetable fields through an incubation experiment with combined inhibition methods. The results showed that the ammonia oxidation process is the dominant contributor to $N_2O$ production at all five sites, accounting for 88–97% of the total $N_2O$ emissions. Regardless of acidic, neutral, or alkaline soil, AOA-driven $N_2O$ emission rates were consistently higher than AOB-driven $N_2O$ emission rates. Both AOA-driven and AOB-driven $N_2O$ emissions exhibited positive correlations with soil pH, with significant increases in soil $N_2O$ production associated with high pH levels. Therefore, general production mechanisms were derived, such that more attention should be paid to AOA-driven $N_2O$ emissions and to vegetable soils with a relatively high pH in the middle and lower reaches of the Yangtze River.

**Keywords:** intensive vegetable production; greenhouse gas; nitrification inhibitor; ammonia-oxidizing archaea (AOA); ammonia-oxidizing bacteria (AOB)

## 1. Introduction

In response to the growing demand for vegetables, driven by population growth, China has witnessed the significant development of its greenhouse vegetable production over the past four decades, with an area exceeding 4 million hectares [1]. The cultivation of vegetables in a greenhouse is a crucial method for intensive farming, with strategic significance in enhancing the efficiency of natural resource utilization and increasing economic returns for farmers. The regions along the middle and lower reaches of the Yangtze River are typical and important for greenhouse vegetable cultivation in China, encompassing 30% of the country's total planted area and over 10% of local traditional croplands [2].

The application rates of nitrogen (N) fertilizer in greenhouse vegetable ecosystems are significantly higher, ranging from three to five times more compared to staple food agricultural fields [2,3]. Excessive N fertilizer not only contributes significantly and directly to elevated nitrous oxide ($N_2O$) emissions, resulting in reduced N-use efficiency [4,5], but also leads to secondary salinization and the acidification of soil [6]. $N_2O$ is mainly

produced after N fertilizer application and by a series of N conversion processes involving autotrophic nitrification, heterotrophic nitrification, nitrobacteria denitrification, chemical denitrification, etc. [7,8]. However, ammonia oxidation, as a necessary step of nitrification, is the main process of $N_2O$ production in greenhouse vegetable fields [9].

Ammonia oxidation is the first and rate-limiting step responsible for nitrification and $N_2O$ emission. The ammonia oxidation process alone may contribute up to 80% of the total $N_2O$ emission [3,10]. Ammonia oxidation involves the conversion of ammonia ($NH_3$) to nitrate ($NO_3^-$) by both ammonia-oxidizing archaea (AOA) and bacteria (AOB) [11]. Previous studies suggested that in the presence of moist soil conditions and ammonium ($NH_4^+$) amendment, AOB are found to be the primary contributor to both ammonia oxidation and $N_2O$ production [12,13]. However, Giguere et al. [14] found that there were no notable disparities in $N_2O$ production between AOA and AOB. The physiologically diverse nature of AOA and AOB, which are influenced by various soil environmental conditions, results in varying degrees of ammonia oxidation [15,16]. These factors can influence the relative contributions of AOA and AOB to $N_2O$ production in soils [17,18]. There were contradictory findings reported [19].

The excessive application of N fertilizer, frequent irrigation, and tillage, as well as elevated soil temperatures within enclosed environments, can alter the community structure, abundance, and function of soil microorganisms [20]. Long-term greenhouse vegetable cultivation was found to cause a notable modification in soil physicochemical properties and functional microorganisms such as AOA and AOB [21–23]. While many studies focused on the changes in ammonia oxidation under various soil conditions [12,19,21], the characteristics of ammonia oxidation and the contribution of $N_2O$ emissions by AOA and AOB in greenhouse vegetables remain unexplored and contradictory. Zhong et al. [2] proposed that greenhouse vegetable cultivation with long-term high-N-input conditions may result in AOB playing a more significant role in ammonia oxidation compared to AOA. However, Duan et al. [24] reported that AOA played a dominant role in driving both ammonia oxidation and $N_2O$ production in greenhouse vegetable systems subjected to elevated soil temperatures. We assumed that the underlying microbial production mechanisms can be derived across a wide selection of vegetable fields in the middle and lower reaches of the Yangtze River.

Here, we conducted a series of inhibitor incubation experiments using long-term (>10 years) greenhouse vegetable soils. The objectives of this study were (1) to examine the corresponding contributions of AOA and AOB to $N_2O$ emission and (2) to explore the environmental factors and microbial mechanisms underlying $N_2O$ production. Thus, our research findings can serve as a valuable reference for implementing $N_2O$ mitigation measures for greenhouse vegetable production in the middle and lower reaches of the Yangtze River.

## 2. Materials and Methods

### 2.1. Soil Collection and Preparations

Five soil samples were gathered from typical greenhouse fields that have long-term (>10 years) conventional vegetable cultivation in the middle and lower reaches of the Yangtze River. All topsoil samples (0–20 cm) were collected post-harvest of vegetable crops between October and December 2021. The five sites are as follows: (1) Anthrosols in Hefei city, Anhui Province (AnHui) (31°54′ N, 117°15′ E); (2) Inceptisols in Huan'an city, Jiangsu Province (JiangSu) (33°53′ N, 119°30′ E); (3) Ultisols in Jiujiang city, Jiangxi Province (JiangXi) (29°36′ N, 115°49′ E); (4) Ultisols in Wuhan city, Hubei Province (HuBei) (30°20′ N, 114°13′ E); (5) Ultisols in Xiangtan city, Hunan Province (HuNan) (27°51′ N, 113°1′ E).

After removal of any visible stones and plant debris, one part of the sample was air-dried for analysis of fundamental physical and chemical properties, while another portion was sieved ≤2 mm and stored at 4 °C until laboratory incubation. The fundamental physical and chemical properties of the soil are displayed in Table 1.

**Table 1.** Properties of greenhouse vegetable soils across five different sites.

| Sites | Sand (%) | Silt (%) | Clay (%) | SOC (g kg$^{-1}$) | TN (g kg$^{-1}$) | C/N Ratio | NH$_4^+$-N (mg kg$^{-1}$) | NO$_3^-$-N (mg kg$^{-1}$) | pH (1:5) |
|---|---|---|---|---|---|---|---|---|---|
| AnHui | 20.4 | 42.8 | 36.8 | 16.60 ± 2.36 b | 0.97 ± 0.12 c | 17.08 ± 0.90 a | 40.56 ± 3.74 b | 60.84 ± 2.27 c | 6.91 ± 0.12 d |
| JiangSu | 43.8 | 40.9 | 15.3 | 21.95 ± 0.44 a | 1.83 ± 0.12 a | 12.00 ± 0.16 b | 66.83 ± 4.99 a | 122.79 ± 4.62 a | 7.43 ± 0.02 c |
| JiangXi | 22.6 | 37.2 | 40.2 | 7.51 ± 0.07 c | 0.53 ± 0.03 d | 14.23 ± 0.84 b | 13.66 ± 1.70 c | 11.26 ± 1.63 d | 4.89 ± 0.05 e |
| HuBei | 42.3 | 34.1 | 23.6 | 9.11 ± 0.23 c | 0.67 ± 0.07 d | 13.78 ± 1.80 b | 35.67 ± 8.50 a | 56.52 ± 3.98 b | 8.23 ± 0.01 a |
| HuNan | 24.2 | 51.5 | 28.3 | 20.61 ± 0.06 a | 1.56 ± 0.09 b | 13.24 ± 0.84 b | 64.69 ± 2.19 b | 85.47 ± 3.69 c | 7.84 ± 0.02 b |

The data shown are the means ± standard errors of three replicates. Different lowercase letters within the same column indicate significant differences among sites. SOC, soil organic carbon; TN, total nitrogen.

### 2.2. Soil Microcosm Construction and Incubation

Previous studies demonstrated that acetylene can effectively inhibit ammonia oxidation by both AOA and AOB, while 1-octyne can specifically inhibit AOB, and 2-phenyl-4,4,5,5-tetramethylimidazoline-1-oxyl 3-oxide (PTIO) selectively inhibits AOA [13,25,26]. A group of inhibitors was implemented to differentiate the impacts of individual AOA or AOB on N$_2$O emissions. According to procedures by Fan et al. [27] and Zhang et al. [28], an incubation trial was conducted for all soil samples as follows: 20 g fresh soils were added to 120 mL serum bottle for preincubation with 45% water-filled pore space (WFPS) at 25 °C for 7 days to revive soil microbial activity. After the end of pre-incubation, each soil sample was added to 2 mL (NH$_4$)$_2$SO$_4$ solution (amount to 150 mg N kg$^{-1}$ soil) and adjusted to 60% WFPS. Each soil sample was exposed to acetylene (0.01% $v/v$), 1-octyne (0.03% $v/v$), 300 µmol/kg PTIO, or no inhibitors, to establish four treatments. All serum bottles were placed at a constant temperature of 25 °C, with 60% WFPS, and incubated in darkness under tightly sealed condition for 21 days. Approximately 35 mL of gas samples were collected from the headspace using a 50 mL plastic syringe at 1, 2, 3, 4, 5, 6, 7, 9, 12, 15, and 21 days after the start of incubation for analysis of N$_2$O concentration. Prior to gas sampling, the serum bottles were opened for 20 min to maintain aerobic conditions, and corresponding inhibitors were added and then sealed for 6 h to allow complete inhibition. Three serum bottles per treatment were destructively sampled after 0, 7, 14, and 21 days to extract soil NH$_4^+$, NO$_2^-$, and NO$_3^-$. On day 21, the soil samples were additionally used for DNA extraction and quantitative PCR analysis.

### 2.3. Soil Analyses and N$_2$O Measurements

Soil pH was determined using 1:5 (weight:volume) ratios of soil with distilled water using a pH meter (PHS-3C, Shanghai, China). The SOC was analyzed by wet digestion with H$_2$SO$_4$–K$_2$Cr$_2$O$_7$, and the TN was determined using semi-micro Kjeldahl digestion using TiO$_2$, CuSO$_4$, and K$_2$SO$_4$ as catalysts. The soil NH$_4^+$-N and NO$_3^-$-N concentrations were measured using the blue colorimetric method and dual-wavelength colorimetric method on soil extracts with 2 mol L$^{-1}$ KCl, using an ultraviolet-visible spectrophotometer (Qinghua UV1800PC, Shanghai, China). The gas samples were analyzed using a gas chromatograph (Agilent 7890A, Shanghai, China) equipped with an electron capture detector (ECD) for N$_2$O analysis [29].

### 2.4. Soil DNA Extraction and qPCR Analysis

The soil DNA was isolated from 0.5 g fresh soil using a Fast DNA SPIN Kit following the guidelines provided by the manufacturer [30]. The concentration and quality of the obtained soil DNA were assessed using a Nanodrop ND-1000 Spectrophotometer (NanoDrop Technologies Inc., Wilmington, DE, USA). The functional genes were amplified by real-time PCR (Applied Biosystems, Foster City, CA, USA) after the DNA extract was diluted and described as the gene copy number per gram of dry soil. Following Harter et al. [31], the primer sets amo19F/CrenamoA616r48x and amoA1F/amoA2R were used to determine the copy numbers of archaeal and bacterial amoA genes, respectively. The amplification was performed in 20 µL reaction mixtures, which included 10 µL of SYBR Green (TaKaRa, Shiga, Japan), 0.2 µL of Rox DYEII, 1 µL of template, 0.4 µL of each primer (10 µmol L$^{-1}$),

and 8 μL of sterile broth. Standard curves were generated with 10-fold serial dilutions of the plasmid DNA.

*2.5. Calculations and Statistical Analysis*

The $N_2O$ production rates were determined by measuring the linear increase in $N_2O$ concentrations in the headspaces at the start (0 h) and the end (6 h) of each sampling day. The cumulative $N_2O$ production was then calculated based on the production rates and incubation time. The contribution of $N_2O$ emission from AOA, AOB, and others were calculated using the following formulas [3,27].

$$N_2O_{AOA} = C_{N2O(no\ inhibitor)} - C_{N2O(1\text{-}octyne)} - C_{N2O(acetylene)} \tag{1}$$

$$N_2O_{AOB} = C_{N2O(no\ inhibitor)} - C_{N2O(PTIO)} - C_{N2O(acetylene)} \tag{2}$$

$$N_2O_{others} = C_{N2O(acetylene)} \tag{3}$$

where $N_2O_{AOA}$ is $N_2O$-emission derived from AOA (ng N $g^{-1}$ dry soil); $N_2O_{AOB}$ is $N_2O$-emission-derived from AOB (ng N $g^{-1}$ dry soil); $N_2O_{others}$ is $N_2O$-emission-derived from others (ng N $g^{-1}$ dry soil); $C_{N2O(no\ inhibitor)}$ is cumulative $N_2O$ emission for no inhibitor treatment (ng N $g^{-1}$ dry soil); $C_{N2O(acetylene)}$ is $N_2O$ cumulative emission for acetylene treatment (ng N $g^{-1}$ dry soil); $C_{N2O(1\text{-}octyne)}$ is cumulative $N_2O$ emission for 1-octyne treatment (ng N $g^{-1}$ dry soil); and $C_{N2O(PTIO)}$ is cumulative $N_2O$ emission for PTIO treatment (ng N $g^{-1}$ dry soil).

Statistical analyses were performed using SPSS 25 (IBM SPSS Statistics, Suzhou, China). An SNK multiple range test was used to determine whether significant differences were found among the treatment means, at a significance level of 0.05. Figure preparation was carried out using Origin 2022 (OriginLab, Northampton, MA, USA).

## 3. Results

### 3.1. N₂O Fluxes

The peak value of the $N_2O$ flux significantly varied among the different soils, while the trends of the $N_2O$ fluxes were similar across these five soils throughout the 21-day incubation period (Figure 1). With the exception of negligible $N_2O$ emissions at the JiangXi site, the $N_2O$ fluxes at the other four sites were concentrated within a 7-day time frame. The highest peak for the $N_2O$ flux occurred on the second day following N addition. Notably, all sites exhibited their highest $N_2O$ fluxes when no inhibitor was applied. Conversely, acetylene inhibitors resulted in minimal-to-no $N_2O$ emissions and, thus, recorded the lowest levels of the $N_2O$ fluxes. The use of 1-octyne and PTIO inhibitors yielded intermediate levels between the treatment with no inhibitor and the acetylene treatment. After 7 days of incubation time, all treatments at all sites displayed consistently low levels of the $N_2O$ flux.

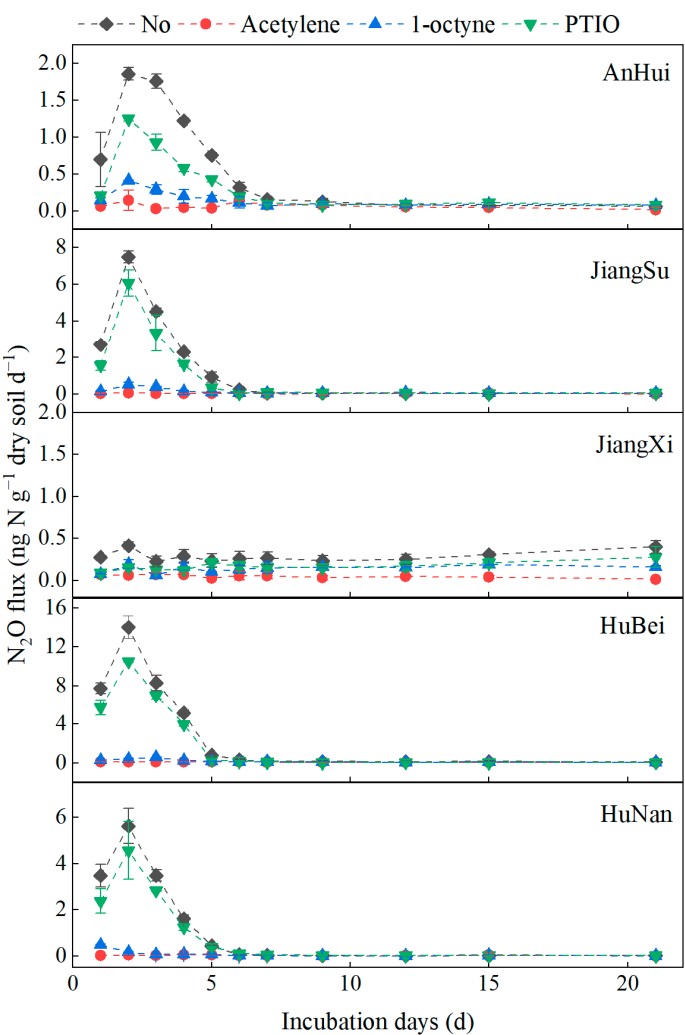

**Figure 1.** Temporal dynamics of soil N$_2$O fluxes by different inhibitors' amendments at different soil sites over a 21-day incubation. Means and standard deviations of triplicate microcosms are plotted.

### 3.2. N$_2$O Emission Derived by AOA and AOB

The cumulative N$_2$O emissions significantly varied among different sites, with the total cumulative N$_2$O emissions ranging from 4.11 to 43.92 ng N g$^{-1}$ dry soil over a 21-day incubation period (Figure 2). The HuBei site exhibited the highest total cumulative N$_2$O emission, 43.92 ng N g$^{-1}$ dry soil, while the JiangSu, HuNan, AnHui, and JiangXi sites showed reductions in cumulative N$_2$O emissions by 53.8%, 62.7%, 83.7%, and 90.6%, respectively, compared to the HuBei site. The N$_2$O emissions driven by different modes exhibited a similar pattern to the total N$_2$O emissions. The highest N$_2$O emissions by AOA or AOB were observed at the HuBei site. However, compared to the HuBei site, the N$_2$O emissions driven by AOA at the JiangSu, HuNan, AnHui, and JiangXi sites decreased by 60.4%, 60.8%, 87.8%, and 94.4%, respectively, while those driven by AOB correspondingly decreased by 54.3%, 70.9%, 79.9%, and 85.1%.

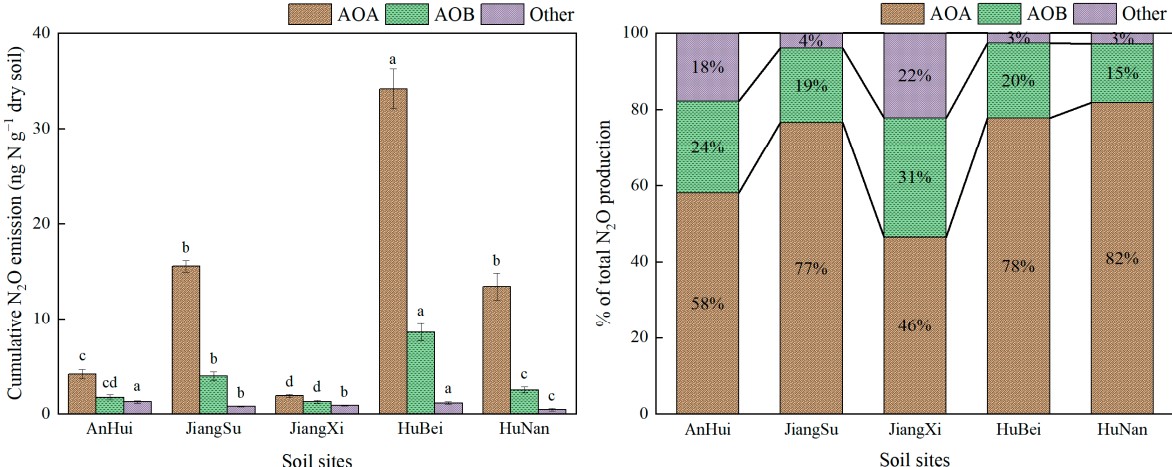

**Figure 2.** The contributions of AOA and AOB to the total $N_2O$ production over a 21-day incubation. Different lowercase letters within the bars indicate significant differences among different soil sites at level of $p < 0.05$.

The $N_2O$ emissions driven by different pathways revealed that the AOA-driven $N_2O$ emissions were the highest across all sites, while other pathways resulted in the lowest emissions (Figure 2). The AOB pathway exhibited intermediate levels of $N_2O$ emissions between AOA and other drivers. The $N_2O$ emissions attributed to AOA accounted for 46–82% of the total $N_2O$ emissions, with AOA-driven $N_2O$ emissions contributing more than 50% of the total emissions when excluding the JiangXi site. The $N_2O$ emissions driven by AOB constituted 15–31% of the total emissions. The $N_2O$ emissions attributed to others contributed to 3–22% of the total emissions, while for sites with high $N_2O$ emissions (HuBei, JiangSu, and HuNan) these other contributions remained below 5%. The $N_2O$ emissions at the three medium or alkaline soil sites (HuBei, JiangSu, and HuNan) exhibited a 1.3–10.7-fold increase compared to those observed at the other two acidic soil sites (AnHui and JiangXi).

### 3.3. Abundance of AOA-amoA and AOB-amoA Genes

In all soil samples, the abundance of AOA-*amoA* functional genes exhibited a significantly higher level compared to the abundance of AOB-*amoA* functional genes (Figure 3). The different additions of inhibitors had a notable impact on the abundance of both AOA-*amoA* and AOB-*amoA* genes. The presence of acetylene and PTIO led to a significant reduction in the abundance of AOA-*amoA* functional genes. The inclusion of acetylene and 1-octyne inhibitors led to a substantial decrease in the abundance of AOB-*amoA* functional genes. Soil samples treated without inhibitors demonstrated elevated levels for both AOA-*amoA* and AOB-*amoA* functional gene abundances across all tested soils. Among the treatments without an inhibitor addition, the HuNan site exhibited the highest level (10.75) of AOA-*amoA* functional gene abundance, whereas the JiangSu, HuBei, AnHui, and JiangXi sites displayed progressively decreasing abundances of the AOA-*amoA* functional genes. Compared with the HuNan site, the abundance of AOA-*amoA* functional genes at the JiangSu, HuBei, AnHui, and JiangXi sites decreased by 3.4%, 4.9%, 16.0%, and 18.7%, respectively. Without the addition of inhibitors, the HuBei site exhibited the highest abundance of AOB-*amoA* functional genes at 10.2, while the HuNan, AnHui, JiangXi, and JiangSu sites showed a successive decrease in AOB-*amoA* functional gene abundance. AOB-*amoA* functional gene abundance decreased by 12.1%, 13.5%, 16.6%, and 28.6% at the HuNan, AnHui, JiangXi, and JiangSu sites, respectively, compared to that at the HuBei site.

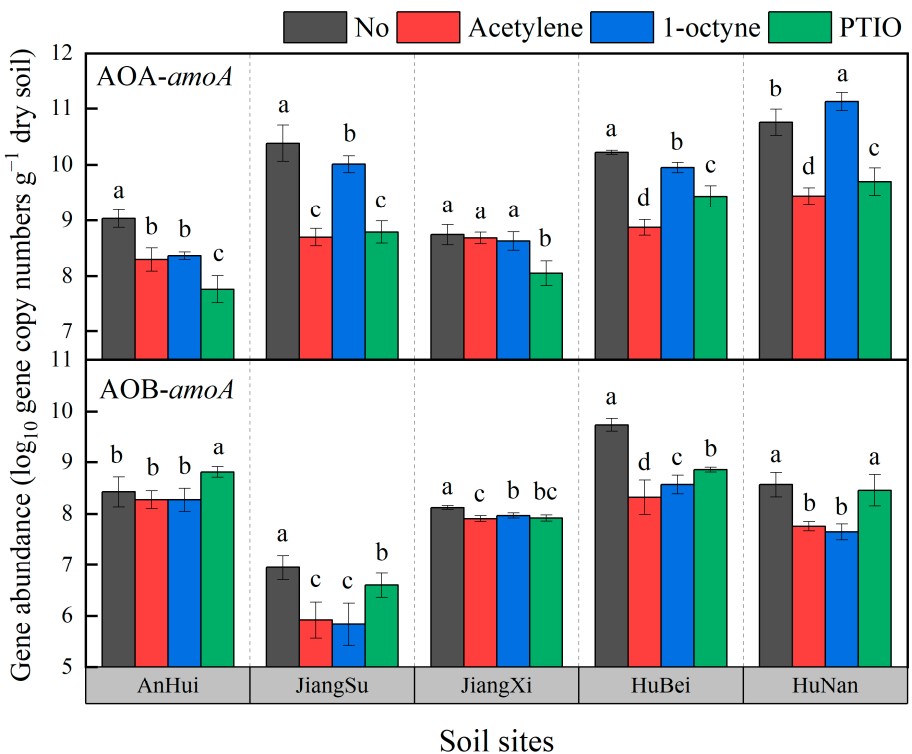

**Figure 3.** The log10-transformed gene numbers for AOA-*amoA* and AOB-*amoA* at different soil sites. Bars indicate the standard deviation (mean ± SD; *n* = 3). Different lowercase letters above the bars indicate significant differences among different inhibitors' amendments at the same site at level of *p* < 0.05.

### 3.4. Correlation between N₂O Production of AOA and AOB and Soil Properties

The correlation analysis revealed a significantly positive association between AOA-driven $N_2O$ emissions and soil pH ($R^2 = 0.59$) as well as between AOB-driven $N_2O$ emissions and soil pH ($R^2 = 0.40$) (Figure 4). No significant correlation was found between the C:N ratio and either AOA-driven or AOB-driven $N_2O$ emissions. Both $NH_4^+$-N and $NO_3^-$-N concentrations exhibited a significantly positive correlation with both AOA-driven and AOB-driven $N_2O$ emissions.

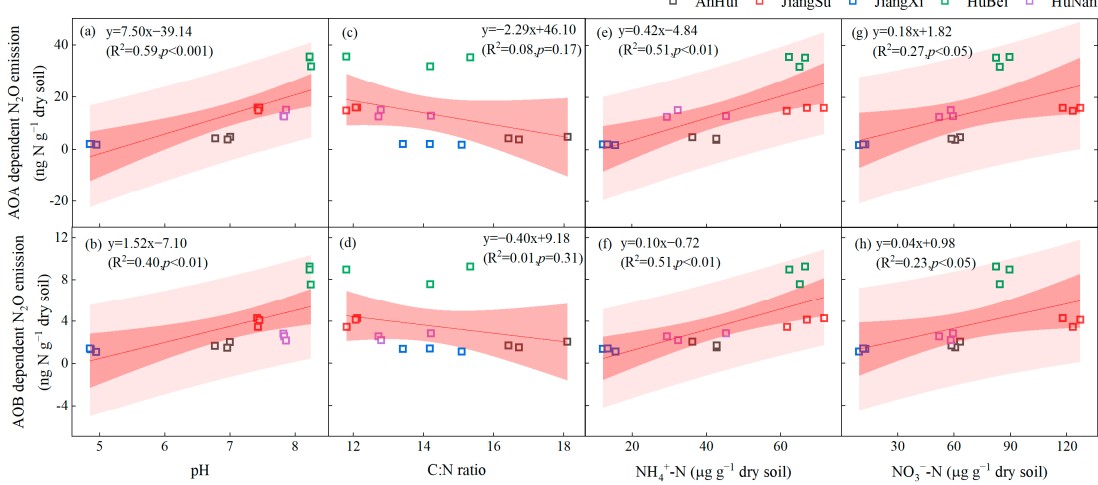

**Figure 4.** The relationship between AOA-driven and AOB-driven $N_2O$ emissions with soil pH (**a,b**), C/N ratio (**c,d**), $NH_4^+$-N (**e,f**), and $NO_3^-$-N (**g,h**) across five field sites. The dark and light red areas represent confidence and prediction intervals at the 95% confidence level, respectively.

## 4. Discussion

### 4.1. Relative Importance of AOA and AOB in N$_2$O Emissions

The negligible N$_2$O fluxes observed with acetylene inhibitors suggested minimal contributions from other processes. Ammonia oxidation was the primary source of N$_2$O emissions, accounting for 78–97% of the total N$_2$O emissions, while the other pathways accounted for only 3–22% of the total N$_2$O emissions in our study (Figure 2). This finding is consistent with previous studies [13,28]. Soil N$_2$O production processes primarily involve autotrophic nitrification, heterotrophic nitrification, nitrobacteria denitrification, and chemical denitrification [7–9]. Excessive N input and frequent tillage in greenhouse vegetable fields provide substrates and aerobic conditions for ammonia-oxidizing microorganisms [32,33]. Moreover, prolonged vegetable cultivation has resulted in the establishment of a distinct and dominant microbial population in the soil, specifically autotrophic nitrification dominated by ammonia-oxidation microorganisms [1].

The N$_2$O emissions at five sites exhibited similar trends, primarily driven by AOA, regardless of acidic, neutral, or weakly alkaline soils (Figure 2). Although this observation was consistent with previous studies [24,34], it was inconsistent with some others stating that AOB predominantly drives N$_2$O emissions in acidic soil, while AOA plays a dominant role in alkaline soil [27,28]. This phenomenon may be attributed to the significant influence of the high-temperature conditions in greenhouse vegetable fields on AOA-driven N$_2$O emissions. Previous studies indicated that the AOA-driven ammonia oxidation process exhibits higher temperature adaptability, with the optimal temperature for AOA significantly exceeding that of AOB [3,34,35]. The vegetable fields were exposed to elevated temperatures in the greenhouse, which could potentially influence the dominant process of soil ammonia oxidation, though our incubation experiment was conducted at a constant temperature of 25 °C. Typically, temperatures in the greenhouse vegetable field reach 50–60 °C during the summer season, and this high-temperature season is usually accompanied by elevated N input. Thus, the field would have shifted in its dominant process for N conversions and microbial community structures under such a long-term planting regime [1]. The results pertaining to microbial functional genes without inhibitors (Figure 3) further support our assertion, revealing a significantly higher abundance of AOA functional genes compared to AOB functional genes, by one–two orders of magnitude. Consequently, for the effective mitigation of N$_2$O emissions, it is imperative to prioritize strategies aimed at reducing AOA-driven N$_2$O emissions from greenhouse vegetable production in this region.

### 4.2. N$_2$O Production Affected by Soil Properties

As substrates and products of nitrification and denitrification, a significant positive relationship was observed between N$_2$O production and soil ammonium and nitrate concentrations (Figure 4). However, these were soil background values when we collected them. We did not measure the dynamic changes during our incubations, which may explain the weak contribution of soil ammonium and nitrate to N$_2$O production in the incubations (Figure 4).

Importantly, we observed a significant positive correlation between both AOA-driven N$_2$O and AOB-driven N$_2$O emissions and soil pH, which was consistent with the findings of Fan et al. [27]. Soil pH is recognized as one of the crucial factors influencing N$_2$O production by AOA and AOB [36,37]. Acidic soil conditions were found to significantly inhibit AOB activity, leading to a reduction in AOB-driven N$_2$O emissions [38]. Conversely, an increase in soil pH creates a favorable environment for AOB bacteria and enhances their microbial activity [31,39]. In our study, it was observed that an increase in soil pH led to a promotion of the abundance of AOB functional genes (Figure 3), as well as an enhancement in AOB-driven N$_2$O emissions (Figure 2). However, with increasing pH levels, there was also a stimulation of AOA-driven N$_2$O emissions (Figure 2). Notably, we found a significant correlation between increased N$_2$O emissions and soil pH from ammonia oxidation in our study. The N$_2$O emissions in the three medium or alkaline soils (HuBei, JiangSu, and HuNan) exhibited a 1.3–10.7-fold increase compared to those

observed in the other two acidic soils (AnHui and JiangXi). These results aligned with previous research on the promotion of the ammonia oxidation process by pH [28]. In Fan et al. [27], the soil C/N ratio was identified as a significant factor influencing AOA- and AOB-driven $N_2O$ emissions. However, our study did not find any correlation between the C/N ratio and $N_2O$ emissions. This discrepancy may be attributed to the relatively consistent C/N ratio in our investigation, though with substantial variations in the TN and SOC among the different soils (Table 1). Given the considerable heterogeneity in soil physical and chemical properties (e.g., soil texture and pH), other factors than the C/N ratio index exerted a greater influence on $N_2O$ emission [40,41]. As pH increased, there was a significant rise in soil $N_2O$ emission. Notably, the $N_2O$ flux trend of the JiangXi soil was obviously different from that of the other four soils. For the JiangXi soil, there was no $N_2O$ flux peak during the 21-day incubation period, regardless of the addition of inhibitors (Figure 1), indicating that the low pH in the JiangXi soil (4.89) would have caused no substantial $N_2O$ emissions even after extra N addition. Soil pH is an important factor affecting ammonia oxidation and $N_2O$ production, with significant positive correlations observed [37,42]. Duan et al. [3] also reported that acidic soil exhibited diminished potential ammonia oxidation and $N_2O$ emission. In our study, the $N_2O$ emissions driven by AOA or AOB showed a significant positive correlation with pH, providing further evidence to support our perspective. Therefore, mitigating $N_2O$ emissions heightened attention toward alkaline soil conditions. Curbing pH levels within alkaline soils presented an effective measure for mitigating $N_2O$ emissions in greenhouse vegetable fields located in the middle and lower reaches of the Yangtze River.

## 5. Conclusions

The incubation research assessed the contributions of AOA and AOB to ammonia oxidation and the resulting $N_2O$ in greenhouse vegetable soils located in the middle and lower reaches of the Yangtze River. Although the $N_2O$ emissions varied among the different soil sites, a consistent production mechanism was observed. The dominant process driving $N_2O$ emissions at all soil sites was ammonia oxidation, while the contributions from other processes were negligible. Regardless of the soil acidity (acidic, neutral, or weakly alkaline), AOA-driven ammonia oxidation rather than AOB-driven ammonia oxidation played a major role in $N_2O$ emissions. Both AOA-driven and AOB-driven $N_2O$ emissions exhibited positive correlations with soil pH, with significant increases in soil $N_2O$ production associated with higher pH levels. Therefore, to mitigate greenhouse gas emissions from vegetable cultivation soils in this region, particular attention should be paid to control AOA-driven $N_2O$ emissions in the middle and lower reaches of the Yangtze River. Furthermore, curbing soil pH levels within alkaline soils presents an effective measure for mitigating $N_2O$ emissions from greenhouse vegetable fields in the middle and lower reaches of the Yangtze River.

**Author Contributions:** Conceptualization, Z.X.; project administration, Y.D.; methodology, X.X.; lab analysis, J.Z.; software, Y.J. and B.W.; writing—original draft, review and editing, Y.D. and Z.X.; visualization, C.W.; supervision, Z.X. All authors have read and agreed to the published version of the manuscript.

**Funding:** This work was jointly supported by the National Natural Science Foundation of China (42377292) and the Jiangsu Province Special Project for Carbon Peak and Carbon Neutral Science and Technology Innovation (BE2022421).

**Data Availability Statement:** No new data were created in this study.

**Conflicts of Interest:** The authors declare no conflict of interest.

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
