# Peer review of "Contributions of Ammonia-Oxidizing Archaea and Bacteria to Nitrous Oxide Production in Intensive Greenhouse Vegetable Fields"

_agronomy, doi:10.3390/agronomy13092420_

Round 1
Reviewer 1 Report
The article named differentiated N2O production mechanisms of ammonia-oxidizing archaea and bacteria across five greenhouse vegetable fields in the middle and lower reaches of Yangtze River presents important information on...to mitigate greenhouse gas emissions from vegetable cultivation soils in China., evaluating N2O emissions from these soils. They found that Soil pH emerged as a crucial factor influencing N2O production and emissions.
The manuscript is well written; however some details have to be checked: Please, add number of lines, check for references cited in text. It is needed to improve the text.
Check for: ng N g-1 of what? soil?
Figures: legends: explain soil sites codes, by extense
Please, check the full text.
the text seems to be too short. too short, but reasonably well written.
Author Response
Thank you for your recognition of our work and constructive comments for improving the manuscript!
We have revised the title, abstract, and other sections to improve the whole manuscript.
We changed all “ng N g-1” to “ng N g-1 dry soil”.
The soil sites are coded after provincial names as shown in the new Figure 1.
Please check the revised version according to the journal style requirement.
We appreciate all your nice comments!
Reviewer 2 Report
The article is interesting because it reports N2O emissions as a function of biological activity in 5 soil classes. N2O is a greenhouse gas and reducing its emissions is one of the global goals.
I consider the topic original and relevant, mainly for identifying the bacteria present in soils. However, there was a flaw in the identification, as it only quantified the microorganisms and did not describe their taxonomy.
Comparing with other published material in the subject area, the research adds the use of Acetylene in inhibiting the activity of ammonia-oxidizing archaea (AOA) and bacteria (AOB) in different soil classes.
The Introduction is well written; doesn't need improvement/extension.
The methodology is overall appropriate, however it needs the correction Soil Taxonomy.
The conclusions are consistent with the evidence and arguments presented, they address the main question. N2O emissions are related to soil acidity and alkalinity.
The references are appropriate.
Tables and figures are adequate. The reading is clear and facilitates the understanding of the author's text. They presented the necessary data to understand the research.
The paper is suitable for publication. However, it needs minor revisions.
Comments are in the text

Author Response
Thank you for your recognition of our work and constructive comments for improving the manuscript!
We have revised the title, abstract, and other sections to improve the whole manuscript. We have supplemented new keywords.
The soil sites are coded after provincial names as shown in the new Figure 1.
We modified the soil classification with reference to the American Soil Taxonomy in the revised version.
Please check the revised version according to the journal style requirement.
We appreciate all your nice comments!
Reviewer 3 Report
The subject matter of the paper is very interesting, but the manner in which the detailed research was carried out raises many concerns. Samples were taken from five locations and this is clear, while the way in which further research was carried out is not so clear. Detailed information regarding laboratory repetitions is missing. Studies for a given location conducted on a single sample cannot be the basis for broader conclusions, as they are not representative. Therefore, the material presented is not sufficient for publication. I also suggest that excessive abbreviations should not be used in the tables. If they are already included, they must be explained below the tables.
The paper lacks a clearly defined research hypothesis and research questions.
The conclusions should answer the research questions posed.
The paper is not prepared according to the guidelines of the journal.
Author Response
Thank you for your recognition of our work and constructive comments for improving the manuscript! The article has been revised as follows:
(1) We added the map information of sampling sites as new Figure 1.
(2) We added the expression of test method in the materials and methods.
(3) We removed the abbreviated codes and replaced with the full site name.
(4) We revised the research hypothesis in abstract and introduction.
(5) We checked and revised reference list style and polished English writing throughout the whole manuscript.